# A Novel Deep Learning Method Based on an Overlapping Time Window Strategy for Brain–Computer Interface-Based Stroke Rehabilitation

**DOI:** 10.3390/brainsci12111502

**Published:** 2022-11-05

**Authors:** Lei Cao, Hailiang Wu, Shugeng Chen, Yilin Dong, Changming Zhu, Jie Jia, Chunjiang Fan

**Affiliations:** 1Department of Artificial Intelligence, Shanghai Maritime University, Shanghai 201306, China; 2Department of Rehabilitation Medicine, Huashan Hospital, Fudan University, Shanghai 200040, China; 3Department of Rehabilitation Medicine, Wuxi Rehabilitation Hospital, Wuxi 214001, China

**Keywords:** brain–computer interface, motor attempt (MA), EEG, deep learning method, overlapping time window

## Abstract

Globally, stroke is a leading cause of death and disability. The classification of motor intentions using brain activity is an important task in the rehabilitation of stroke patients using brain–computer interfaces (BCIs). This paper presents a new method for model training in EEG-based BCI rehabilitation by using overlapping time windows. For this aim, three different models, a convolutional neural network (CNN), graph isomorphism network (GIN), and long short-term memory (LSTM), are used for performing the classification task of motor attempt (MA). We conducted several experiments with different time window lengths, and the results showed that the deep learning approach based on overlapping time windows achieved improvements in classification accuracy, with the LSTM combined vote-counting strategy (VS) having achieved the highest average classification accuracy of 90.3% when the window size was 70. The results verified that the overlapping time window strategy is useful for increasing the efficiency of BCI rehabilitation.

## 1. Introduction

Stroke leads to high rates of disability and death worldwide [1]. To restore brain function affected by stroke, patients need to undergo rigorous rehabilitation. Currently, there are a variety of approaches to help restore motor function after a stroke, including the use of mirror therapy [2], virtual reality [3], aerobic exercise [4], and brain–computer interface (BCI) technology [5,6,7]. BCI technology can help patients recover independently and perform tasks by efficiently controlling additional devices, making it a good option for patients.

The use of a non-invasive BCI for motor rehabilitation has become a focus of current research and is considered a mainstream experimental method. Patients perform motor imagery (MI) or motor attempt (MA) tasks based on cues from the system. The BCI then decodes and converts the motor intents from the electroencephalogram (EEG) signals into commands and provides feedback according to the experimental protocol [8,9]. However, EEG signals may be unstable or random and show significant individual differences. In addition, EEG signals intended for the same behavior but collected at different times and under different circumstances may also have large differences; hence, it is difficult to classify EEG signals directly. Feature extraction and classification algorithms are needed to extract meaningful information from the multidimensional EEG signals [10].

Based on previous studies, the features extracted by traditional machine learning methods can be classified into three categories: spatial, time, and frequency. In dichotomous BCI tasks, the common spatial pattern (CSP) algorithm is the most common method for extracting spatial features [11], as it is able to extract the spatially distributed components of each class from multichannel EEG signals. Many studies have expanded the CSP algorithm, and the filter bank CSP (FBCSP) developed from the CSP algorithm has achieved very good classification performance in MI-BCI [12]. Analyzing EEG signals in a time series can yield rich statistical features. Geethanjali et al. extracted seven time-domain features from EEG signals and classified them using linear discriminant analysis [13]. As many EEG signal features are reflected in the frequency domain, analysis of frequency domain features is important for BCIs. Furthermore, by converting EEG signals from the time domain to the frequency domain, the distribution and variation of EEG frequencies can be visualized. Chen et al. visualized event-related synchronization and event-related desynchronization in MA and MI tasks in different frequency bands [14]. Although the above method can be applied to MA-BCI and MI-BCI to some extent, they require prior knowledge and manually designed features combined with the use of machine learning for classification, which may present problems of insufficient feature extraction and low adaptability to different patients. Hence, many studies have tried to use deep learning (DL) to automatically learn features gathered from EEG signals for classification.

In contrast to traditional machine learning methods, DL does not require predefined feature vectors, as it can automatically learn latent and highly abstract features from raw EEG signals. The combination of BCI with DL methods has been used in the rehabilitation of patients. Lin et al. developed a convolutional neural network (CNN)-based model for predicting BCI rehabilitation outcomes [15]. Liang et al. used the long short-term memory (LSTM) neural network for generating motor trajectories of the lower-extremity exoskeleton for stroke rehabilitation [16] and a graph embedding-based model, Ego-CNN, for identifying key graph structures during MI [17]. However, BCI systems using DL methods require large amounts of EEG data for training models, which results in a bottleneck in therapy. At present, independent patients experience more difficulty performing control tasks due to tedious experimental steps, which often leads to less data collected and less than optimal accuracy for classification using DL methods.

For smaller datasets, data augmentation techniques have proven to be an effective way to improve the performance of DL models, and the approach is to generate more data from the original data for training the model. In previous studies, some have performed data enhancement by adding noise to the original EEG signals [18]. Sliding time windows are advantageous in augmenting EEG signal data. Hartmann et al. used an overlapping time window to expand a dataset of epileptic patients [19], and Zhang et al. extracted time and frequency domain features from multiple windows for a classification task of left versus right hand movements [20]. In addition, several studies have used generative adversarial network models for generating new data similar to the original EEG signal [21]. This approach can help to compensate for the inability to collect large amounts of EEG signals from patients during motor rehabilitation and further improve the performance of DL methods for classifying EEG signals.

The purpose of this paper was to improve the performance of DL on MA-BCI through data augmentation techniques to contribute to the rehabilitation training of patients. Specifically, we provide more accurate neurofeedback by improving the recognition accuracy of a patient’s motor intention. To achieve this aim, we propose a DL method based on overlapping time windows for the classification tasks of MA-BCI. This study compares the classification performance of three different DL models on MA tasks. To investigate the effect of different time periods on BCI classification, we visualized the classification results on a time series and analyzed the differences in EEG signals at different time slices using the power spectral density topography of the brain.

## 2. Materials and Methods

The data used in our experiment were collected from 7 stroke subjects using BCI interventions. Demographic information and clinical data are reported in Table 1. All subjects typically performed three sessions per week; one session included ninety trials, and each trial corresponded to one type of task: motor attempt (MA) or idle state (IS).

### 2.1. Experimental Protocol

Figure 1 shows the experimental protocol during rehabilitation training. The experimental setup consists of two components: a BCI module and a force feedback device. The BCI module is responsible for the collection and analysis of EEG signals, and the force feedback device is responsible for providing neurofeedback. The patient’s stroke-affected hand was immobilized on the force feedback device that was controlled by the BCI system. When the experimental task was a motor attempt, patients continually attempted wrist extension with the affected hand. When the experimental task was in an idle state, they were ordered to rest and do nothing. The force feedback device drove the patient’s stroke-affected hand to complete a wrist extension movement when the BCI system accurately identified the patient’s motor intention. For incorrect identification, the device would stay stationary.

### 2.2. Data Acquisition and Preprocessing

According to Figure 1, the EEG signals of each trial were recorded for 11 seconds and started with a white arrow image used to prompt the patient to be prepared. Three seconds later, a task cue (red geometrical shapes) was displayed on the screen, and the patient was asked to perform either a movement attempt or a rest state. After the cue disappeared, the patient was told to continue performing the task following the cue until the white cross disappeared. Then, the patients rested for 1.5 s. The recorded signals were sampled by a 32-channel EEG cap, and the EEG electrodes were placed according to the international 10–20 system. The sampling frequency was 200 Hz. Data from 31 channels were used for calculation, and the filter range was 4 to 40 Hz. Some examples of the preprocessed EEG signals (C3, C4) from the motor function areas of the brain are shown in Figure 2. Five seconds of EEG before the white cross disappeared from each trial were extracted for training the model.

### 2.3. Overlapping Time Window

The performance of DL models is very heavily dependent on the quantity of data involved in the training. Due to the difficulty of collecting MA data, the amount of data collected for individual patients is small. In existing work, a promising approach is to split the individual signals into multiple subsignals for training the model [19,20,22]. We propose a data augmentation method based on an overlapping time window for increasing the number of instances during training. The raw EEG data were segmented by overlapping windows; each data window served as an independent instance. The number of windows was controlled by two parameters: the time window length *L* and the overlap rate *O*. For the original input of experimental data $X^{i}=\left[x_{1},x_{2},\ldots ,x_{T}\right]\in R^{C^{*}T}$, *i* represented the type of task, *C* represented the channel, and *T* represented the sampling point; in this experiment, *C* = 31 and *T* = 1000. Given the parameters *O*, *L*. The raw data were segmented into $D_{L,O}^{i}$.
(1)$$D_{L,O}^{i}=\left\{X_{1,}^{i}X_{2}^{i},\ldots ,X_{s,}^{i}\ldots X_{n}^{i}\right\}\in R^{n^{*}C^{*}L}$$
(2)$$X_{s}^{i}=\left\{x_{t},x_{t+1},\ldots ,x_{t+L}|t=1+\left(s-1\right)LO\right\}$$

$n=\frac{T\;-\;L}{LO}+1$, which denotes that the original data were sliced into *n* time segments. The segmented data and the original data had the same task label *i*. When n=1, the data were not segmented. We divided the original dataset into a training set and a test set. For the training set, each signal $X_{train}$ with a length of 1000 was divided into 32 windows (*L* = 60, *O* = 0.5). In this way, the signals in each window were used as instances to train the model. In the testing phase, for each signal $X_{test}$ with a length of 1000 in the test dataset, we divided it into 32 windows using the same approach. The data from these windows were fed into the trained model, and the classification results were obtained on these windows. After that, multiple window classification results of windows were fused into one decision for $X_{test}$ by using the vote-counting strategy (VS). In addition, we also designed another method for classifying $X_{test}$. We combined the features of different windows from the last hidden layer of the model by summing, and the combined feature was fed into the softmax layer for classification, which is called the feature fusion strategy (FFS) in this paper. “&VS" and “&FFS” refer to strategies for validating trained models on the test set using voting and feature fusion, respectively.

### 2.4. Graph Isomorphism Network Model

#### 2.4.1. Graph Data Construction

The aim of graph neural networks (GNNs) is to use graph structure data and node features as input to learn a representation of the node (or graph) for relevant tasks [23]. Because EEG data are easily converted to graph structure data, several studies have investigated GNNs applied to EEG signal-based tasks [17,24,25,26,27]. An important aspect of using a GNN to classify EEG signals is building graph data, the original data first need to be converted into graph structure data. The EEG signal of a window can be defined as G=(V,E), where *V* and *E* represent the sets of nodes and edges, respectively. In this experiment, we treated the individual channel as a node and the closed channels as connected edges. Specifically, the average Euclidean distance *d* between the CZ channel and the other channels was calculated, and the two channels with an electrode distance less than *d* were treated as connected.

#### 2.4.2. Graph Isomorphism Network

Most GNNs complete the graph classification process through a strategy of aggregating information from neighbors. Formally, node updates and the graph embedding $h_{G}$ are obtained using the following formula.
(3)$$a_{v}^{\left(k\right)}=\mathrm{AGGREGATE}{}^{\left(k\right)}\left(\left\{h_{u}^{\left(k-1\right)}|u\in N\left(v\right)\right\}\right)$$
(4)$$h_{v}^{\left(k\right)}={\mathrm{COMBINE}}^{\left(k\right)}\left(h_{v}^{\left(k-1\right)},a_{v}^{\left(k\right)}\right)$$
(5)$$h_{G}=\mathrm{READOUT}\left(\left\{h_{v}^{\left(K\right)}|v\in G\right\}\right)$$
where $h_{v}^{\left(k\right)}$ is the feature vector of node *v* at the *k* -th iteration layer, $h_{v}^{\left(0\right)}$ represents node input, and N(v) is a set of nodes adjacent to *v*. AGGREGATE and COMBINE represent the aggregation of information about neighbors and the aggregation of information about oneself and neighbors, respectively. The choices of ${\mathrm{AGGREGATE}}^{\left(k\right)}\left(\cdot \right)$ and ${\mathrm{COMBINE}}^{\left(k\right)}\left(\cdot \right)$ in GNNs are crucial. The output $h_{G}$ aggregates node features via the READOUT function at the final iteration.

In this study, we used a graph isomorphism network (GIN) model for classifying EEG signals. The GIN network is a kind of GNN and uses the summation method to complete AGGREGATE, COMBINE, and READOUT [28]. Following the literature [28], we used the following formula for feature updates of node features:(6)$$h_{v}^{\left(k\right)}={\mathrm{MLP}}^{\left(k\right)}\left(\left(1+\epsilon^{\left(k\right)}\right)\cdot h_{v}^{\left(k-1\right)}+\sum_{u\in N(v)}h_{u}^{\left(k-1\right)}\right)$$

MLP represents multilayer perceptrons, and ϵ is a parameter that can be trained. We tuned the hyperparameters through a grid search over the training set in this experiment. The search space ranges for the network depth and the number of neurons in GIN were defined as {1, 2, 3, 4} and {32, 64, 128, 256}, respectively. After hyperparameter optimization, the network depth *k* and the number of neurons were set to 2 and 256, respectively. The learned graph embedding $h_{G}$ passes through two fully connected layers to output the final feature representation. This experiment focused on the binary classification problem, so the number of neurons in the final fully connected layer was 2. In this study, the output layers of the three models were the softmax layer, and the loss functions were all set to the cross-entropy loss function.

### 2.5. CNN

Convolutional neural networks (CNNs) are considered to be one of the most successful deep learning models and have been widely used for feature extraction of EEG signals [24,29,30]. The CNN is a deep feed-forward neural network that includes crucial convolutional operations. Compared to traditional neural networks, CNNs reduce the training parameters by local sensing and weight sharing. Each convolutional layer consists of multiple convolutional kernels of the same size for feature extraction. The mathematical description of the convolutional operation is as follows:(7)$$y_{mn}=f\left(\sum_{j=0}^{J-1}\sum_{i=0}^{I-1}x_{m+i,n+j}w_{ij}+b\right)$$
where *x* represents the matrix on which the convolution operation is performed, and *y* is the output of the convolution. *I*, *J* corresponds to the size of the convolution kernel *w*, *b* represents a bias, and *f* is the activation function, which was ReLu in this study. The grid search ranges of the parameters were defined as follows: number of convolution layers and max-pooling layers {1, 2, 3, 4}, length of the convolutional kernels {2, 3, 4, 5}, number of convolutional kernels {16, 32, 64, 128}, and number of neurons in the fully connected layer {32, 64, 128, 256}. The optimized model structure in this study consisted of 3 convolutional layers and 3 max pooling layers. The size of the convolutional kernels was 4 * 4, 2 * 2, and 2 * 2, and the number of convolutional kernels was 32, 64, and 128, respectively. After completing the pooling of the final layer, we flattened the extracted features and fed them into a fully connected layer $fc_{1}$, which contained 128 neurons. Finally, the output of the fully connected layer $fc_{1}$ passed through the ReLu activation function and another fully connected layer $fc_{2}$ to output the final 2-dimensional representation feature.

### 2.6. LSTM

Due to the long duration of the patient performing the task, some useful features still needed to be retained despite the long interval. LSTM can retain the motor intention of EEG signals that are both long and short. LSTM networks are a modified version of recurrent neural networks (RNNs) [31]. Based on RNNs, the LSTM added a multiple gate structure (forget gate $f_{t}$, input gate $i_{t}$, and output gate $o_{t}$) for updating the cell state. The LSTM network layer contains the cell state $C_{t}$, which represents the cell information stored at time *t*. The data features $x_{t}$ at time *t*, the hidden features $h_{t-1}$ at moment t−1, and the cell state $C_{t-1}$, were fed into the LSTM nodes, which were processed by gates to output the hidden state and cell state at the next moment. The calculations are as follows.
(8)$$f_{t}=\mathrm{sigmoid}\left(w_{f}\left[x_{t},h_{t-1}\right]+b_{f}\right)$$
(9)$$i_{t}=\mathrm{sigmoid}\left(w_{i}\left[h_{t-1},x_{t}\right]+b_{t}\right)$$
(10)$${\bar{C}}_{t}=\tanh\left(w_{c}\left[h_{t-1},x_{t}\right]+b_{c}\right)$$
(11)$$C_{t}=f_{t}*C_{t-1}+i_{t}*{\bar{C}}_{t}$$
where $w_{f}$ and $b_{f}$ indicate the weight and bias of the forget gate, respectively. The sigmoid function in the forget gate determines which messages need to be deleted. The corresponding input gate $i_{t}$ determines which information to retain, and $w_{i}$ and $b_{i}$ indicate the weight and bias of the input gate. ${\bar{C}}_{t}$ represents the candidate hidden state, and $w_{c}$ and $b_{c}$ correspond to the weight and bias, respectively. The output of the forget gate and the input gate are jointly calculated to obtain the cell state value $C_{t}$ at the current moment.

Finally, the current cell state $C_{t}$ and the output $o_{t}$ of the output gate are calculated as follows to obtain the current hidden state $h_{t}$.
(12)$$o_{t}=\mathrm{sigmoid}\left(w_{o}\left[h_{t-1},x_{t}\right]+b_{o}\right)$$
(13)$$h_{t}=o_{t}*\tanh\left(C_{t}\right)$$
where $w_{o}$ and $b_{o}$ are the weight and bias of the output gate, respectively. For a multilayer LSTM model, the hidden state $h_{t}$ at moment *t* of the previous layer is used as the input of the next network layer at moment *t*. The number of LSTM hidden layer units was determined by the time window length, and each time point was a unit. In this study, we employed a two-layer LSTM, and we fed the hidden state at the last moment of the last layer into a fully connected layer to output the final feature representation. The hidden state features of the LSTM perform a grid search in the range {32, 64, 128, 256}, with an optimized feature size of 128.

### 2.7. Evaluation Procedures

One of the most important aspects of BCI is accuracy. To test the effectiveness of different methods, we used 3-fold cross-validation. For one session, the data were randomly divided into a training set containing 60 trials and a test set containing 30 trials, with a ratio of 2:1. We first optimized hyperparameters on the training set via grid search, with 90% of the data used to train the model and 10% to validate the performance of the hyperparameters and choose the model structure with the highest average accuracy. The data ratio between the two task categories was always 1:1 in the different sets. After completing hyperparameter optimization, we trained the final classification models using all the training data. The average accuracy of each fold was used to evaluate the performance of the model. In addition, we used the information transfer rate (ITR) to evaluate the performance of the BCI [32]. The units of ITR are bits/min, which are calculated from Equation (Equation 14). *N* is the number of task types, which is set to 2 in this study, *P* is the accuracy rate, and *T* is the time during the task (60 s).
(14)$$B=\left\{\log_{2}N+P\log_{2}P+\left(1-P\right)\log_{2}\left[\frac{1-P}{N-1}\right]\right\}\times \frac{60}{T}$$

## 3. Results and Discussion

### 3.1. Overall Performance

In this study, the average classification results for seven subjects are reported in Table 2. The results listed include six methods and the accuracies achieved with the same segmentation strategy (L = 60, O = 0.5). The results showed that the LSTM&FFS achieved the highest mean accuracy of 90.1% for all subjects, while the GIN&VS had the lowest accuracy of 81.9%. One of the explanations for differences in accuracy between different methods could be different types of extracted features. To verify that our method is superior to the existing method, we compared our methods with the traditional algorithms [14,33]. After comparison, compared with CSP and FBCSP, the six methods in this study yielded an average improvement of 11.71% and 23.01% in regard to average accuracy. The results indicated that the methods can learn distinctive features of multiple windows for classification and that these features improved classifier performance. To analyze the impact of age on the algorithm, patients were divided into two age groups based on the median age (40 years). Compared to the group aged <40, the group aged ≥40 showed higher accuracy on different algorithms, and the results suggest that the patient’s age may be a factor in the accuracy of the classification.

To identify the performance of different models, we recorded the training loss and validation accuracy during model training for Subject 1 in Figure 3. The loss in the CNN and LSTM training set dropped to 0.2 after 100 iterations and the top accuracy of the test sets converged to approximately 93% and 98%, respectively. A faster convergence was observed in GIN, but the accuracy was relatively low. Figure 4 illustrates the training time and the number of training parameters for different models. Although the training parameters of the LSTM model were not the highest, training of the model required more time. This may partially be because of the structure of the LSTM model [34], which could not complete parallel computing in the training process. For real-world applications, the choice of method can consider multiple factors of accuracy and computational complexity.

Paired-sample t-tests were used to determine whether the difference among individual methods was statistically significant. The results are presented in Table 3. Significant differences were found across multiple methods at the 0.05 significance level, and the difference between LSTM&VS and LSTM&FFS was not significant at the 0.01 significance level. Although the performance was different in the six methods, the overall accuracy was high.

### 3.2. Effects of Window Size

To investigate the effect of window size, we experimented with multiple window sizes and evaluated the performances of the models in the corresponding window. The sizes of the comparison time windows were 1000, 200, 100, 90, 80, 70, and 60. Figure 5 presents the average accuracy of the six methods for the different windows. The lowest accuracy was achieved when the window size was 1000, which means without data augmentation. The most likely reason for this phenomenon is probably because the models overfit fewer training data. We illustrate the performance of each method for three window sizes in Table 4. The LSTM&VS accuracy was 90.3% when the window size was 70. However, the LSTM model showed only a classification accuracy of 65.4% without using a time window. The difference in accuracies occurred because the different window sizes changed the lengths of the input time series. In addition, according to the results, when the window size was 70, the average accuracy of each method was higher than that of the method with a window size of 100. This may indicate that relatively small sizes of windows had better performance.

### 3.3. Generic Performance of BCI

It is difficult to compare different BCI systems since there are many aspects that can influence the performance of BCI, such as input, preprocessing, and outputs. The ITR is a widely and generally accepted standard by which the performance of different BCI systems can be compared [35]. Figure 6 illustrates the distribution of ITRs for the sessions. The average ITR for all seven subjects was 10.72 ± 4.82 bits/min. Several subjects (1, 3, 4 and 7) reached the highest ITR with 12 bits/min. Subject 7 had the lowest ITR of 1.73 bits/min. For the ITR of motor attempts, fewer results have been reported. Khalaf et al. obtained an average ITR of 40.83 bits/min for a four-class task [36]. Zeng et al. achieved the highest ITR of 24 bits/min during ankle rehabilitation robot training [37]. In this study, the value of ITR is negatively correlated with *T*, and the value of ITR is limited by task time.

### 3.4. The Visualization of Feature Distribution

To investigate the validity of the time window. A visualization technique called TSNE [38] was used to downscale the learned features for visualization. Figure 7 shows the distribution of features for different time windows. The different colored scatter points in the figure indicate the different task types, and each scatter represents the extracted feature by one window. We observed that the features extracted by methods with time windows were easier to classify. After training, multiple EEG signals of time segments were well identified. However, some EEG signals from different windows were hard to identify. In addition, it was also found that among CNN, GIN, and LSTM, LSTM performed best in feature extraction, which had fewer segments that could not be distinguished. To further observe the LSTM performance, we constructed the LSTM&VS confusion matrices of the seven subjects in Figure 8. The correct classification accuracies are shown on the diagonal cells. It can be seen that the LSTM&VS accuracy in each task was similar for individual subjects. The results demonstrated that LSTM&VS obtained good overall performance.

### 3.5. The Impact of the Number of Network Layers

The number of network layers usually affected the model performance. Table 5 shows the accuracy of models with different numbers of network layers. Compared with other settings, the accuracy of 1 layer was lower. In addition, the accuracies of the CNN&VS and CNN&FFS were more likely to be influenced by the number of network layers, while the accuracies of LSTM&VS and LSTM&FFS were more stable.

### 3.6. The Visualization of Accuracy on Time Window

In this section, to analyze the classification differences in each time window, we conducted statistics and visualized the classification results of the EEG signals on the windows by using the LSTM&VS method. Figure 9 shows the sequence of classification results. Each row represents a session, and each column represents the classification statistics of the time window, in which the time windows with higher accuracy are highlighted. As seen in the figure, the distribution of accuracy across the time windows differs in patients. For Subject 3, the time window with higher classification accuracy appeared in the first window, and the window with lower classification accuracy appeared in the last window. The difference in sequence probably occurred because the appearances of the discriminative motor intentions were random during the MA experiment.

### 3.7. Study of Cortical Activity on the Time Window

To further investigate the differences in the EEG signals of time segments with different classification accuracies, we used power spectral density topography to represent the frequency domain information of brain signals. Figure 10 illustrates the topography of alpha power for two tasks. Based on the classification accuracy of the time segments in Figure 9, four different time windows were selected for visualization. For the motor attempt task, the PSD of patients was higher in the frontal lobe. Several studies have indicated that stroke can affect the brain function in the frontal lobe [39,40]. The observations may suggest that motor attempts of patients were associated with the frontal cortical regions, which is consistent with a previous study [41]. Channel information from frontal regions may be important for identifying the brain’s motor intentions. When the EEG signals in the frontal lobe are not significant enough, it may contribute to lower classification accuracy.

### 3.8. Limitations in Current Work

The findings in this study are limited by the quantity of data collected, and it is difficult to determine the quantity of data that can be classified well without using a time window. In addition, the optimal filter band and model hyperparameters were not selected according to the subjects, which may limit the ability of the models in different patients. In future work, we will investigate the use of feature engineering for reducing the dimensionality of the model inputs. In addition, the fixed starting point for time window sampling may reduce the performance of the BCI system [42]. Therefore, we will optimize the set of time windows by using a window selection algorithm in the feature work.

## 4. Conclusions

This study showed that for classification tasks during BCI-based stroke rehabilitation, deep learning algorithms based on overlapping time windows achieved good accuracy. It may support improvements in the performance of brain–computer interfaces to generate accurate neurofeedback. One of the more significant findings to emerge from this study is that the distribution of classification results differed across the time windows of the subjects, and it means that there is a possibility of improving classification performance by choosing different windows for classification for different subjects. Therefore, future work can expand on the selection of the time window.

## Figures and Tables

**Figure 1 brainsci-12-01502-f001:**
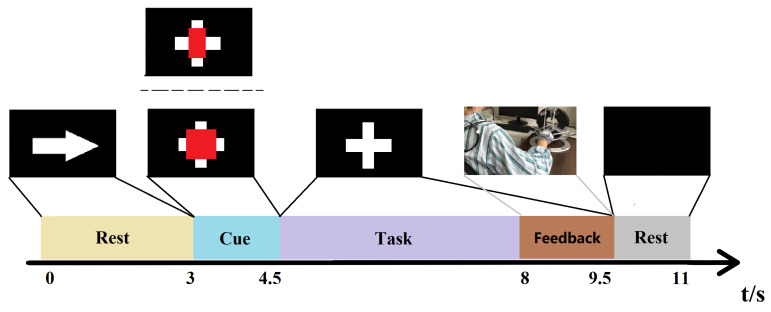
Experimental protocol of rehabilitation training. During the cueing period, a red rectangle is used to alert the user to perform specific tasks. When the cue is a red square, the patient will attempts wrist extension using the stroke-affected hand as hard as possible until the white cross disappears. When the cue is a red rectangle, the patient just needs to stay rested. The patient’s stroke-affected hand was passively extended by the force feedback device when the system accurately identified the patient’s motor intention.

**Figure 2 brainsci-12-01502-f002:**
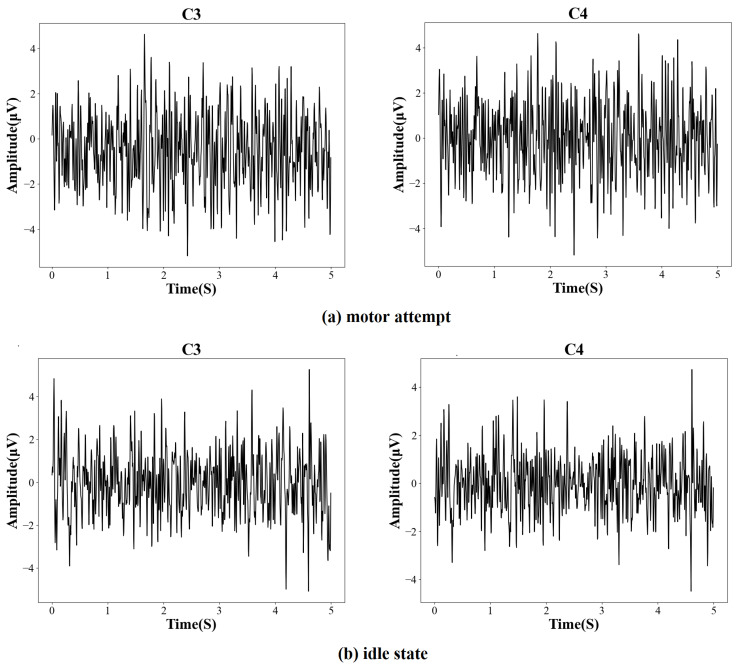
Examples of preprocessed EEG signals from different brain activities.

**Figure 3 brainsci-12-01502-f003:**
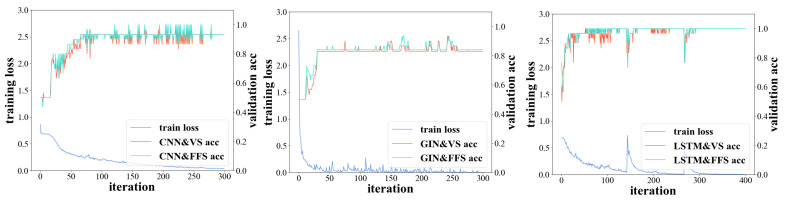
Losses of the training sets for deep learning models and classification accuracies of test sets for different methods.

**Figure 4 brainsci-12-01502-f004:**
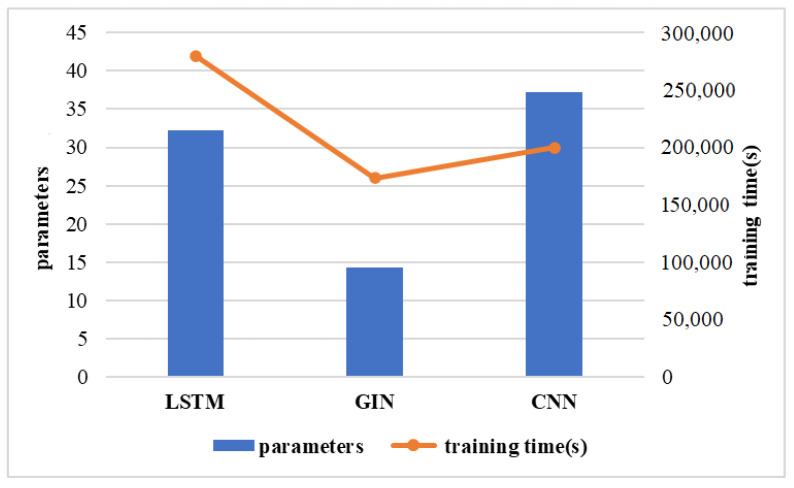
The number of parameters and running times for different models.

**Figure 5 brainsci-12-01502-f005:**
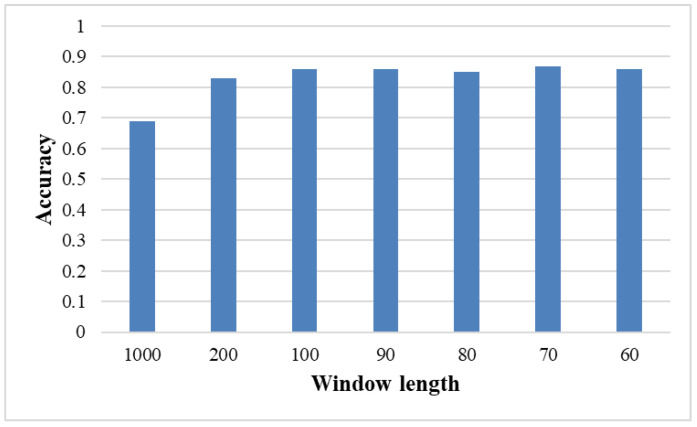
Accuracies for all methods with different window lengths, each column represents the average of the six method accuracies.

**Figure 6 brainsci-12-01502-f006:**
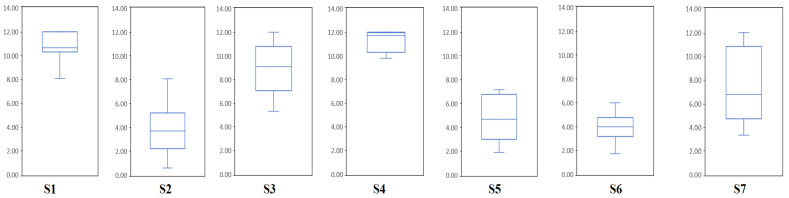
Boxplots represent the ITRs of different sessions for each subject. Each box plot includes 12 sessions of data. The upper and lower lines represent the maximum and minimum ITRs, respectively. The lines in the boxplot represent the median ITR.

**Figure 7 brainsci-12-01502-f007:**
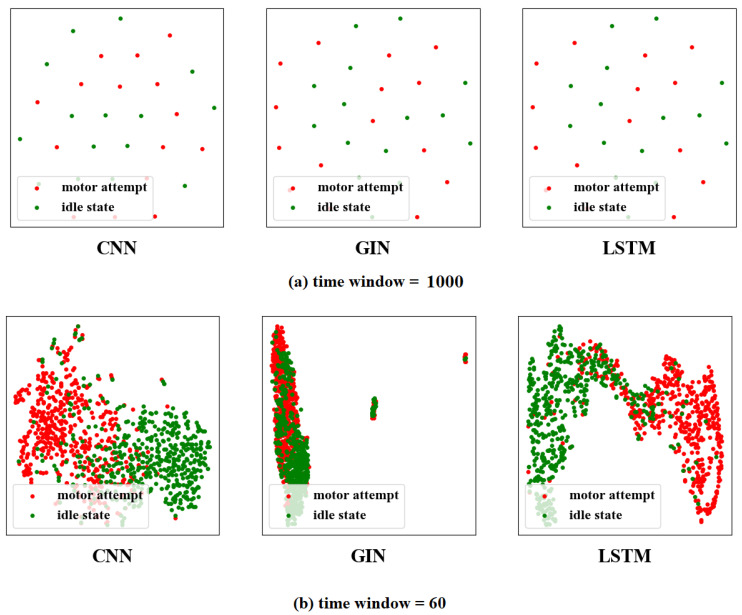
Feature visualization of different models for Subject 1. For time window (**a**), each scatter represents a feature extracted on one trail, and for time window (**b**), each scatter represents a feature extracted on one time window of one trail.

**Figure 8 brainsci-12-01502-f008:**
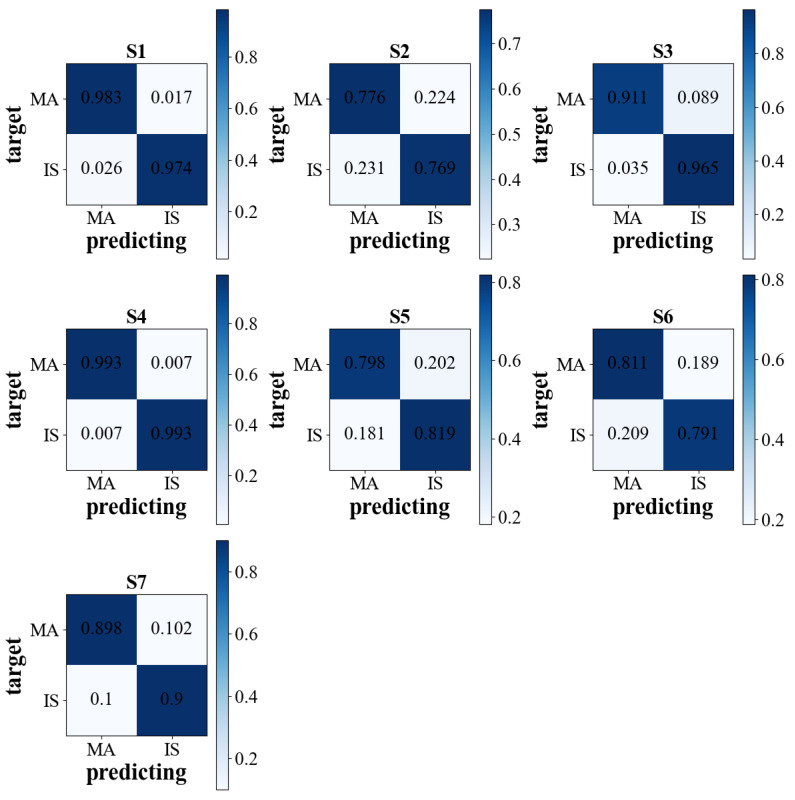
The confusion matrices for all subjects with the LSTM&VS method.

**Figure 9 brainsci-12-01502-f009:**
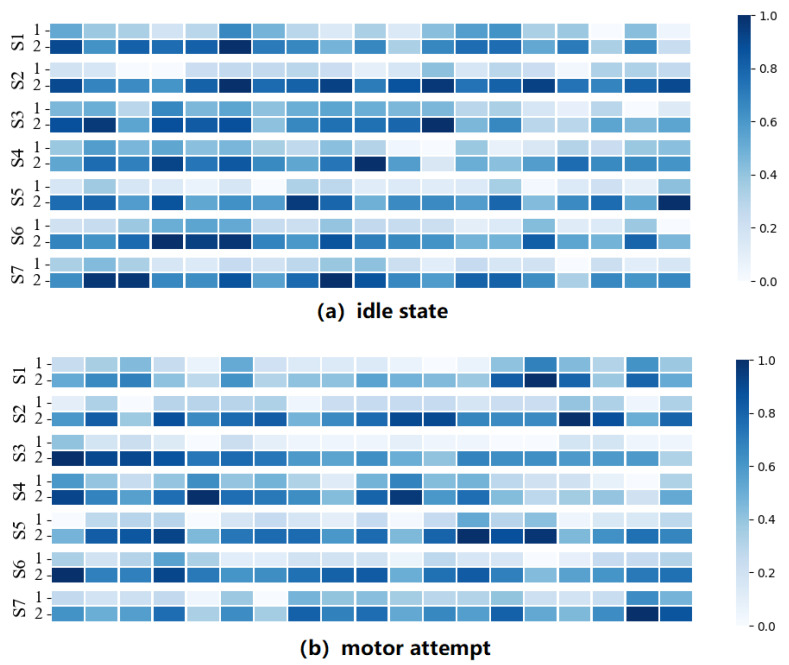
Visualization of classification results learned by LSTM&VS. In each state, two different sessions (1, 2) of seven subjects are visualized. Each row represents a session and each square represents the correct result for classification in a time window.

**Figure 10 brainsci-12-01502-f010:**
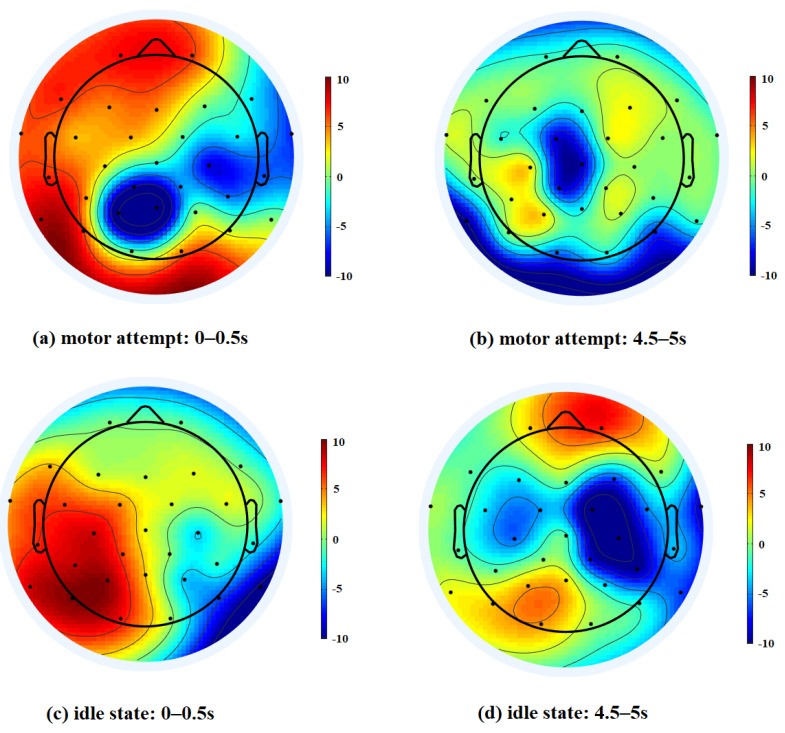
The visualization of the averaged topography of the PSD over the alpha band in different time windows. Session 3 with two task types for Subject 3 is visualized.

**Table 1 brainsci-12-01502-t001:** Demographic information of the subjects.

Subject	Sex	Age	Affected Limb	Stroke Stage
Sub1	Male	31	Right	Subacute
Sub2	Male	40	Left	Subacute
Sub3	Male	42	Right	Subacute
Sub4	Male	47	Right	Subacute
Sub5	Male	36	Right	Subacute
Sub6	Male	30	Right	Subacute
Sub7	Male	65	Left	Subacute
Mean	-	41.6 ± 12.0	-	-

**Table 2 brainsci-12-01502-t002:** The overall comparison results of average classification performance.

Subjects	Accuracy							
	CSP	FBCSP	GIN&VS	LSTM&VS	CNN&VS	GIN&FFS	LSTM&FFS	CNN&FFS
sub1	0.798 ± 0.079	0.614 ± 0.096	0.914 ± 0.061	0.979 ± 0.025	0.944 ± 0.059	0.931 ± 0.049	0.980 ± 0.017	0.934 ± 0.058
sub2	0.700 ± 0.064	0.578 ± 0.096	0.722 ± 0.069	0.772 ± 0.106	0.741 ± 0.065	0.746 ± 0.060	0.806 ± 0.111	0.757 ± 0.057
sub3	0.771 ± 0.077	0.627 ± 0.094	0.902 ± 0.075	0.938 ± 0.044	0.932 ± 0.047	0.919 ± 0.073	0.949 ± 0.042	0.940 ± 0.046
sub4	0.804 ± 0.072	0.635 ± 0.157	0.938 ± 0.038	0.993 ± 0.011	0.966 ± 0.024	0.952 ± 0.030	0.994 ± 0.010	0.970 ± 0.016
sub5	0.724 ± 0.054	0.644 ± 0.057	0.711 ± 0.066	0.808 ± 0.071	0.762 ± 0.076	0.757 ± 0.073	0.841 ± 0.068	0.769 ± 0.085
sub6	0.687 ± 0.040	0.630 ± 0.053	0.744 ± 0.055	0.801 ± 0.037	0.764 ± 0.058	0.777 ± 0.055	0.825 ± 0.029	0.755 ± 0.050
sub7	0.707 ± 0.082	0.677 ± 0.090	0.798 ± 0.078	0.899 ± 0.066	0.845 ± 0.085	0.842 ± 0.081	0.909 ± 0.062	0.856 ± 0.084
Mean	0.742	0.629	0.819	0.884	0.851	0.846	0.901	0.854
Group1	0.736 ± 0.056	0.629 ± 0.015	0.790 ± 0.108	0.865 ± 0.099	0.823 ± 0.105	0.822 ± 0.095	0.884 ± 0.088	0.820 ± 0.100
Group2	0.746 ± 0.050	0.630 ± 0.041	0.840 ± 0.099	0.901 ± 0.094	0.871 ± 0.101	0.865 ± 0.092	0.915 ± 0.080	0.881 ± 0.096

Group1, age < 40; Group2, age ≥ 40.

**Table 3 brainsci-12-01502-t003:** Paired-sample *t*-test results of different methods.

Method	Comparsion Method	T	df	Sig. (2-Tailed)
GIN&VS	GIN&FFS	−5.451	6	0.002
	CNN&FFS	−5.345	6	0.002
	LSTM&FFS	−7.528	6	<0.001
	CNN&VS	−6.705	6	0.001
	LSTM&VS	−7.140	6	<0.001
CNN&VS	GIN&FFS	1.146	6	0.295
	CNN&FFS	−0.999	6	0.356
	LSTM&FFS	−5.976	6	0.001
	LSTM&VS	−5.805	6	0.001
LSTM&VS	GIN&FFS	6.823	6	<0.001
	CNN&FFS	4.333	6	0.005
	LSTM&FFS	−3.527	6	0.012
GIN&FFS	CNN&FFS	−1.501	6	0.184
	LSTM&FFS	−8.477	6	<0.001
CNN&FFS	LSTM&FFS	−5.400	6	0.002

**Table 4 brainsci-12-01502-t004:** Comparison of the accuracies of different methods with time windows of 60, 70, and 1000.

Window Length	Method	Accuracy							
		Sub1	Sub2	Sub3	Sub4	Sub5	Sub6	Sub7	Mean
70	GIN&VS	0.929 ± 0.061	0.764 ± 0.069	0.915 ± 0.075	0.942 ± 0.038	0.760 ± 0.066	0.781 ± 0.055	0.858 ± 0.078	0.850
	GIN&FFS	0.924 ± 0.049	0.755 ± 0.060	0.912 ± 0.073	0.947 ± 0.030	0.755 ± 0.073	0.782 ± 0.054	0.843 ± 0.081	0.845
	CNN&VS	0.94 ± 0.039	0.766 ± 0.076	0.931 ± 0.047	0.967 ± 0.021	0.777 ± 0.059	0.789 ± 0.054	0.871 ± 0.083	0.864
	CNN&FFS	0.939 ± 0.040	0.754 ± 0.079	0.929 ± 0.049	0.960 ± 0.018	0.761 ± 0.053	0.760 ± 0.059	0.856 ± 0.095	0.851
	LSTM&VS	0.985 ± 0.018	0.808 ± 0.101	0.951 ± 0.041	0.997 ± 0.008	0.844 ± 0.072	0.819 ± 0.036	0.917 ± 0.066	0.903
	LSTM&FFS	0.984 ± 0.014	0.801 ± 0.101	0.948 ± 0.038	0.995 ± 0.008	0.842 ± 0.078	0.812 ± 0.047	0.917 ± 0.078	0.900
100	GIN&VS	0.925 ± 0.046	0.764 ± 0.058	0.916 ± 0.065	0.929 ± 0.039	0.746 ± 0.056	0.793 ± 0.053	0.844 ± 0.074	0.845
	GIN&FFS	0.919 ± 0.053	0.755 ± 0.048	0.919 ± 0.069	0.935 ± 0.039	0.745 ± 0.070	0.784 ± 0.056	0.846 ± 0.080	0.843
	CNN&VS	0.938 ± 0.058	0.759 ± 0.057	0.923 ± 0.047	0.941 ± 0.030	0.755 ± 0.052	0.762 ± 0.053	0.852 ± 0.075	0.847
	CNN&FFS	0.925 ± 0.056	0.744 ± 0.064	0.920 ± 0.055	0.946 ± 0.029	0.748 ± 0.053	0.734 ± 0.057	0.830 ± 0.096	0.835
	LSTM&VS	0.982 ± 0.020	0.799 ± 0.095	0.943 ± 0.039	0.995 ± 0.008	0.807 ± 0.077	0.893 ± 0.038	0.816 ± 0.078	0.891
	LSTM&FFS	0.986 ± 0.017	0.792 ± 0.098	0.942 ± 0.040	0.995 ± 0.010	0.814 ± 0.084	0.898 ± 0.046	0.815 ± 0.089	0.892
1000	GIN	0.810 ± 0.057	0.732 ± 0.062	0.814 ± 0.091	0.797 ± 0.051	0.696 ± 0.067	0.700 ± 0.037	0.731 ± 0.057	0.754
	CNN	0.689 ± 0.084	0.625 ± 0.058	0.763 ± 0.013	0.643 ± 0.066	0.634 ± 0.051	0.612 ± 0.062	0.638 ± 0.067	0.658
	LSTM	0.674 ± 0.061	0.630 ± 0.051	0.719 ± 0.111	0.633 ± 0.075	0.644 ± 0.042	0.629 ± 0.042	0.646 ± 0.064	0.654

**Table 5 brainsci-12-01502-t005:** Accuracy comparison between different numbers of network layers.

Conv Layers	Accuracy					
	GIN&VS	CNN&VS	LSTM&VS	GIN&FFS	CNN&FFS	LSTM&FFS
1 layer	0.803 ± 0.082	0.821 ± 0.091	0.882 ± 0.079	0.834 ± 0.074	0.791 ± 0.072	0.900 ± 0.065
2 layers	**0.819 ± 0.085**	0.844 ± 0.086	**0.884 ± 0.078**	**0.846 ± 0.075**	0.846 ± 0.068	**0.901 ± 0.066**
3 layers	0.816 ± 0.078	**0.851 ± 0.083**	0.883 ± 0.068	0.843 ± 0.082	**0.854 ± 0.080**	0.896 ± 0.069
4 layers	0.806 ± 0.089	0.844 ± 0.088	0.882 ± 0.079	0.821 ± 0.078	0.847 ± 0.069	0.900 ± 0.072
Mean	0.810	0.840	0.882	0.836	0.835	0.900

The highest classification accuracy for a given method are bold marked.

## Data Availability

The data presented in this study are available on request from the corresponding author.

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
