# Peer review of "A Novel Deep Learning Method Based on an Overlapping Time Window Strategy for Brain–Computer Interface-Based Stroke Rehabilitation"

_brainsci, 2022, doi:10.3390/brainsci12111502_

Round 1

Reviewer 1 Report

In this manuscript, some typical deep neural networks using overlapping time windows were used for BCI-based stroke rehabilitation. The topic in interesting and the writing seems good, however, there are some issues that should be addressed.

(1) Please better explain how you "rehabilitate" the stroke patients using your proposed method. Do you apply a neurofeedback? If so, what is the role of your proposed method in applying the neurofeedback?

(2) There are many parameters and hyperparameters for designing different deep neural networks. Please explain in detail how you set those parameters, e.g.

k=2 and number of neurons and layers in GNN

The number of convolutional and max-pooling layers, the size and number of convolutional kernels, the number of neurons in fully connected layer in CNN, etc.

(3) When you compare different models, the structure of them all should be optimized. If not, the comparison would not be valid. Did you do such optimizations?

(4) Please report both mean and standard deviation of the performance criteria wherever applicable.

(5) In page 6, 0.93% and 0.98% should be replaced by 93% and 98%.

(6) In Figure 5, what is the similarity of the same sessions between different subjects that encourage you to show them with the same color to be comparable? E.g. why should we compare sessions 1 of different subjects together and not with sessions 2? 

(7) Why did you visualize the PSD distribution of just 9Hz as the representative of alpha band and not that of the whole alpha band (e.g. integration of the whole alpha band)?

Author Response

Dear Reviewer,

Thank you for your letter and for the reviewers’ comments concerning our manuscript. Those comments are all valuable and very helpful for revising and improving our paper, as well as the important guiding significance to our researches. We have studied comments carefully and have made correction which we hope meet with approval. The main corrections in the paper and the responds (in BOLD type) to the reviewer’s comments are as flowing:

1. Please better explain how you "rehabilitate" the stroke patients using your proposed method. Do you apply a neurofeedback? If so, what is the role of your proposed method in applying the neurofeedback?

Response: As the reviewer suggested, We had complemented the explanation of the introduction of the rehabilitation. And we had applied neurofeedback, and the role of our proposed method in applying the neurofeedback is to decide whether the force device performs neurofeedback. The related description had been added to this manuscript.

 1)“The purpose of this paper was to improve the performance of DL on MA-BCI through data augmentation techniques to contribute to the rehabilitation training of patients. Specifically, we provide more accurate neurofeedback by improving the recognition accuracy of a patient's motor intention.”  (Section of Introduction, Page 2)

2)“Figure 1 shows the experimental protocol during rehabilitation training. The experimental setup consists of two components: a BCI module and a force feedback device. The BCI module is responsible for the collection and analysis of EEG signals, and the force feedback device is responsible for providing neurofeedback. The patient's stroke-affected hand was immobilized on the force feedback device that was controlled by the BCI system. When the experimental task was a motor attempt, patients continually attempted wrist extension with affected hand. When the experimental task was in an idle state, they were ordered to rest and do nothing. The force feedback device drove the patient's stroke-affected hand to complete a wrist extension movement when the BCI system accurately identified the patient's motor intention. For incorrect identification, the device would stay stationary. ” (Section of Materials and Methods, Page 3)

3)“It may support improvements in the performance of brain-computer interfaces to generate accurate neurofeedback.” ( Section of Conclusion, Page 15)

2. There are many parameters and hyperparameters for designing different deep neural networks. Please explain in detail how you set those parameters, e.g.k=2 and number of neurons and layers in GNN.The number of convolutional and max-pooling layers, the size and number of convolutional kernels, the number of neurons in fully connected layer in CNN, etc.

Response:We tuned the hyperparameters by grid searching over the training set in this experiment. And we had introduced detailed performing in the manuscript. 

1)“The search space ranges for the network depth and the number of neurons in GIN were defined as {1, 2 , 3, 4} and {32, 64, 128, 256}, respectively. After hyperparameter optimization, the network depth k and the number of neurons were set to 2 and 256, respectively.(Section of Materials and Methods, Page 6)

2)“The grid search ranges of the parameters were defined as follows: number of convolution layers and max-pooling layers {1, 2, 3, 4}, length of the convolutional kernels {2, 3, 4, 5}, number of convolutional kernels {16, 32, 64, 128}, and number of neurons in the fully connected layer {32, 64, 128, 256}. The optimized model structure in this study consisted of 3 convolutional layers and 3 max pooling layers. The size of the convolutional kernels was 4*4, 2*2, and 2*2, and the number of convolutional kernels was 32, 64, and 128, respectively.”  (Section of Materials and Methods, Page 6)

3)“The number of LSTM hidden layer units was determined by the time window length, and each time point was a unit. In this study, we employed a two-layer LSTM, and we fed the hidden state at the last moment of the last layer into a fully connected layer to output the final feature representation. The hidden state features of the LSTM perform a grid search in the range {32, 64, 128, 256}, with an optimized feature size of 128.”  (Section of Materials and Methods, Page 7)

4)“We first optimized hyperparameters on the training set via grid search, with 90% of the data used to train the model and 10% to validate the performance of the hyperparameters and choose the model structure with the highest average accuracy.” (Section of Materials and Methods, Page 7)

3. When you compare different models, the structure of them all should be optimized. If not, the comparison would not be valid. Did you do such optimizations?

Response: We were sorry about that negligence of the introduction of optimization of model structures what were used for comparison. We complemented the model structure optimizations in the manuscript.

“We first optimized hyperparameters on the training set via grid search, with 90% of the data used to train the model and 10% to validate the performance of the hyperparameters and choose the model structure with the highest average accuracy.” (Section of Materials and Methods, Page 7)

4. Please report both mean and standard deviation of the performance criteria wherever applicable.

Response: As the reviewer suggested that. We had updated Table 2, Table 4, and Table 5. (Section of Results and Discussion, Page 8, Page 10 and Page 12)

5. In page 6, 0.93% and 0.98% should be replaced by 93% and 98%.

Response: We were sorry about these mistakes and we had corrected them in the manuscript.

“The loss in the CNN and LSTM training set dropped to 0.2 after 100 iterations and the top accuracy of the validation sets converged to approximately 93% and 98%, respectively.(Section of Results and Discussion, Page 8)

6. Figure 5, what is the similarity of the same sessions between different subjects that encourage you to show them with the same color to be comparable? E.g. why should we compare sessions 1 of different subjects together and not with sessions 2? 

Response: We were very sorry that this was a misunderstanding due to our negligence.  Here we wish to compare the maximum and minimum values of ITR on sessions for different subjects. Considering the misunderstandings that may result, we have modified Figure 6 to the boxplots and modified the parts of the manuscript that relate to it.

Figure 6. Boxplots represent the ITRs of different sessions for each subject. Each box plot includes 12 sessions of data. The upper and lower lines represent the maximum and minimum ITRs, respectively. The lines in the boxplot represent the median ITR. (Figure 6. Page 11)

7. Why did you visualize the PSD distribution of just 9Hz as the representative of alpha band and not that of the whole alpha band (e.g. integration of the whole alpha band)?

Response: We were sorry about that the using of the PSD distribution of 9Hz as the representative of alpha band were wrong, and the PSD distribution should be discussed on the whole alpha band. Hence, we recounted the power distribution over the whole alpha band and updated Figure 11. The related description had been added to this manuscript.

1) “Figure 10. The visualization of the averaged topography of the PSD over the alpha band in different time windows. Session 3 with two task types for Subject 3 is visualized.” (Figure 10. Page 14)

2)“Figure 10 illustrates the topography of alpha power for two tasks. Based on the classification accuracy of the time segments in Figure 9, four different time windows were selected for visualization. For the motor attempt task, the PSD of patients was higher in the frontal lobe. Several studies have indicated that stroke can affect brain function in the frontal lobe [39,40]. The observations may suggest that motor attempts of patients were associated with the frontal cortical regions, which is consistent with a previous study [41].  Channel information from frontal regions may be important for identifying the brain's motor intentions. When the EEG signals in the frontal lobe are not significant enough, it may contribute to lower classification accuracy.” (Section of Results and Discussion, Page 12-13)

Reviewer 2 Report

1. In figure 1, can you explain what's the procedure difference between motor attempt and idle? Just by looking at the experimental paradigm, there is no much difference from the pictures.

2. What are the demographic information of the subjects such as age group, male, female, would those factors have impact on the result, is the algorithm sensitive to different age groups?

3. In the dataset section, can you show some example EEG signals for different brain activities?

4. How is test and train data divided, what's the ratio? 

5. Figure 2, there are two methods, but the training loss looks only one line?

6. It is better to switch the label order for the parameters and the training time given the order of the y-axis.

Author Response

Dear Reviewer,

Thank you for your letter and for the reviewers’ comments concerning our manuscript. Those comments are all valuable and very helpful for revising and improving our paper, as well as the important guiding significance to our researches. We have studied comments carefully and have made correction which we hope meet with approval. The main corrections in the paper and the responds (in BOLD type) to the reviewer’s comments are as flowing:

1. In figure 1, can you explain what's the procedure difference between motor attempt and idle? Just by looking at the experimental paradigm, there is no much difference from the pictures.

Response:We were sorry that the the procedure difference between motor attempt and idle had not been introduced. We had modified Figure1 and rewritten the legends.

Figure 1. Experimental protocol of rehabilitation training. During the cueing period, a red rectangle is used to alert the user to perform specific tasks. When the cue is a red square, the patient will attempts wrist extension using the stroke-affected hand as hard as possible until the white cross disappears. When the cue is a red rectangle, the patient just needs to stay rested. The patient's stroke-affected hand was passively extended by the force feedback device when the system accurately identified the patient's motor intention. (Figure 1, Page  3)

2. What are the demographic information such as age group, male, female, would those factors have impact on the result, is the algorithm sensitive to different age groups?

Response: We were sorry that the demographic information of the subjects had not been introduced and the factor of demographic information should be discussed. We had complemented the detail of the demographic information in Table 1, and the related discussion had been added to this manuscript.

“ To analyse the impact of age on the algorithm, patients were divided into two age groups based on the median age (40 years). Compared to the group aged < 40, the group aged ≥ 40 showed higher accuracy on different algorithms, and the results suggest that the patient's age may be a factor in the accuracy of the classification.”  (Section of Results and Discussion, Page 8)

3. In the dataset section, can you show some example EEG signals for different brain activities?

Response: As the reviewer suggested that, we had add some example EEG signals for different brain activities. And the results are shown in the Figure 2.

4. How is test and train data divided, what's the ratio? 

Response: We were sorry that the division between the training and test sets, as well as their ratios, had not been introduced. Hence, we had complemented the explanation related to the division of training and test sets.   

“For one session, the data were randomly divided into a training set containing 60 trials and a test set containing 30 trials, with a ratio of 2:1.   (Section of Materials and Methods, Page 7)

5. Figure 2, there are two methods, but the training loss looks only one line?

Response: We were very sorry that this was a misunderstanding due to our negligence. There was only one line of training loss in Figure 2 because these 2 methods used the same model for classification. The difference between these strategies was that FFS method used hidden features for pattern recognizing on different time windows, and the other one used the model for label predicting. We complemented the explanations of these two methods and updated Figure 2. Furthermore, we had marked these strategies by different abbreviations in the manuscript.

“ After that, multiple window classification results of windows were fused into one decision for Xtest by using a vote-couting strategy (VS). In addition, we also designed another method for classifying Xtest. We combined the features of different windows from the last hidden layer of the model by summing, and the combined feature was fed into the softmax layer for classification, which is called the feature fusion strategy (FFS) in this paper. "&VS" and "&FFS" refer to strategies for validating trained models on the test set using voting and feature fusion, respectively.”  (Section of Materials and Methods, Page 5)

6. It is better to switch the label order for the parameters and the training time given the order of the y-axis.

Response: As the reviewer suggested that, we had updated Figure 5.

Reviewer 3 Report

By utilising overlapping time windows, the authors demonstrated a new technique for model training in EEG-based BCI rehabilitation. They employed three distinct models for the categorization task of motor attempts: convolutional neural network (CNN), graph isomorphism Network (GIN), and long short-term memory (LSTM) (MA). The authors demonstrated that the overlapping time windows-based deep learning strategy improved classification accuracy, with the LSTM achieving the best average classification accuracy of 90.3% when the window size was 70.Their findings demonstrated the value of the overlapping time window method for boosting BCI rehabilitation effectiveness. Their concept looks fresh and has been executed flawlessly technically. However, the paper has to be proofread because it has a number of grammatical errors. The abstract is well written, the themes are meaningful, insightful, and applicable to future research. The paper is really well written.

Author Response

1. However, the paper has to be proofread because it has a number of grammatical errors. The abstract is well written, the themes are meaningful, insightful, and applicable to future research. The paper is really well written.

Dear Reviewer,

We apologize for the grammatical errors of our manuscript. We worked on the manuscript for a long time and the repeated addition and removal of sentences and sections obviously led to some grammatical errors . We have now worked on both language and readability and have also involved native English speakers for language corrections. We really hope that the flow and language level have been substantially improved for reading. 

Round 2

Reviewer 1 Report

Thanks to authors for addressing all my comments properly.